# $p$-Topologicalness—A Relative Topologicalness in ⊤-Convergence Spaces

## Lingqiang Li

Department of Mathematics, Liaocheng University, Liaocheng 252059, China; lilingqiang0614@126.com;
Tel.: +86-0635-8239926

**Abstract:** In this paper, $p$-topologicalness (a relative topologicalness) in ⊤-convergence spaces are studied through two equivalent approaches. One approach generalizes the Fischer's diagonal condition, the other approach extends the Gähler's neighborhood condition. Then the relationships between $p$-topologicalness in ⊤-convergence spaces and $p$-topologicalness in stratified $L$-generalized convergence spaces are established. Furthermore, the lower and upper $p$-topological modifications in ⊤-convergence spaces are also defined and discussed. In particular, it is proved that the lower (resp., upper) $p$-topological modification behaves reasonably well relative to final (resp., initial) structures.

**Keywords:** fuzzy topology; fuzzy convergence; lattice-valued convergence; ⊤-convergence space; relative topologicalness; $p$-topologcalness; diagonal condition; neighborhood condition

---

## 1. Introduction

The theory of convergence spaces [1] is natural extension of the theory of topological spaces. The topologicalness is important in the theory of convergence spaces since it mainly researches the condition of a convergence space to be a topological space. Generally, two equivalent approaches are used to characterize the topologicalness in convergence spaces. One approach is stated by the well-known Fischer's diagonal condition [2], the other approach is stated by Gähler's neighborhood condition [3]. In [4], by considering a pair of convergence spaces $(X, p)$ and $(X, q)$, Wilde and Kent investigated a kind of relative topologicalness, called $p$-topologicalness. When $p = q$, $p$-topologicalness is equivalent to topologicalness in convergence spaces. They also defined and discussed the lower and upper $p$-topological modifications in convergence spaces. Precisely, for a pair of convergence spaces $(X, p)$ and $(X, q)$, the lower (resp., upper) $p$-topological modification of $(X, q)$ is defined as the finest (resp., coarsest) $p$-topological convergence space which is coarser (resp., finer) than $(X, q)$. Similarly, a topological modification of $(X, q)$ is defined as the finest topological convergence space which is coarser than $(X, q)$.

Lattice-valued convergence spaces are common extension of convergence spaces and lattice-valued topological spaces. It should be pined out that lattice-valued convergence spaces are established on the basis of fuzzy sets. However, the lattice structure is used to replace the unit interval $[0, 1]$ as the truth table for membership degrees. In recent years, two kinds of lattice-valued convergence spaces received much attention: (1) the theory of stratified $L$-generalized convergence spaces based on $L$-filters, which is initiated by Jäger [5] and then developed by many researchers [6–25]; and (2) the theory of ⊤-convergence spaces based on ⊤-filters, which is investigated by Fang [26] in 2017. The topologicalness in stratified $L$-generalized convergence spaces was studied by Jäger [27–29] and Li [30,31], the $p$-topologicalness and $p$-topological modifications in stratified $L$-generalized convergence spaces were discussed by Li [32,33].

The topologicalness in $\top$-convergence spaces was researched by Fang [26] and Li [34]. In this paper, we shall consider the $p$-topologicalness and $p$-topological modifications in $\top$-convergence spaces.

The contents are arranged as follows. Section 2 recalls some basic notions as preliminary. Section 3 discusses the $p$-topologicalness in $\top$-convergence spaces by generalized Fischer's diagonal condition and generalized Gähler's neighborhood condition, respectively. Then the relationships between $p$-topologicalness in $\top$-convergence spaces and $p$-topologicalness in stratified $L$-generalized convergence spaces are established. Section 4 focuses on $p$-topological modifications in $\top$-convergence spaces. The lower and upper $p$-topological modifications in $\top$-convergence spaces are defined and discussed. Particularly, it is proved that the lower (resp., upper) $p$-topological modification behaves reasonably well relative to final (resp., initial) structures.

## 2. Preliminaries

Let $L$ be a complete lattice with the top element $\top$ and the bottom element $\bot$. For a commutative quantale, we mean a pair $(L, *)$ such that $*$ is a commutative semigroup operation on $L$ with the condition

$$\forall a \in L, \forall \{b_j\}_{j \in J} \subseteq L, a * \bigvee_{j \in J} b_j = \bigvee_{j \in J} (a * b_j).$$

$(L, *)$ is called integral if the top element $\top$ is the unique unit, i.e., $\forall a \in L, \top * a = a$. For any $a \in L$, each function $a * (-) : L \longrightarrow L$ has a right adjoint $a \to (-) : L \longrightarrow L$ defined as $a \to b = \bigvee \{c \in L : a * c \leq b\}$. In the following, we list the usual properties of $*$ and $\to$ [35].

(1) $a \to b = \top \Leftrightarrow a \leq b$;
(2) $a * b \leq c \Leftrightarrow b \leq a \to c$;
(3) $a * (a \to b) \leq b$;
(4) $a \to (b \to c) = (a * b) \to c$;
(5) $(\bigvee_{j \in J} a_j) \to b = \bigwedge_{j \in J} (a_j \to b)$;
(6) $a \to (\bigwedge_{j \in J} b_j) = \bigwedge_{j \in J} (a \to b_j)$.

We call $(L, *)$ to be a meet continuous lattice if the complete lattice $L$ is meet continuous [36], that is, $(L, \leq)$ satisfies the distributive law: $a \wedge (\bigvee_{i \in I} b_i) = \bigvee_{i \in I} (a \wedge b_i)$, for any $a \in L$ and any directed subsets $\{b_i | i \in I\}$ in $L$.

$L$ is said to be continuous if $(L, \leq)$ is a continuous lattice [36], that is, for any nonempty family $\{a_{j,k} | j \in J, k \in K(j)\}$ in $L$ with $\{a_{j,k} | k \in K(j)\}$ is directed for all $j \in J$, the identity

$$(\textbf{DD}) \qquad \bigwedge_{j \in J} \bigvee_{k \in K(j)} a_{j,k} = \bigvee_{h \in N} \bigwedge_{j \in J} a_{j,h(j)}$$

holds, where $N$ is the set of all choice functions on $J$ with values $h(j) \in K(j)$ for all $j \in J$. Obviously, continuity implies meet-continuity for $L$.

In this article, unless otherwise stated, we always assume that $L = (L, *)$ is a commutative, integral, and meet continuous quantale.

A function $\mu : X \to L$ is called an $L$-fuzzy set in $X$, and all $L$-fuzzy sets in $X$ is denoted as $L^X$. The operations $\vee, \wedge, *, \to$ on $L$ can be translated pointwisely onto $L^X$. Said precisely, for any $\mu, \nu \in L^X$ and any $\{\mu_t | t \in T\} \subseteq L^X$,

$$\mu \leq \nu \text{ iff } \mu(x) \leq \nu(x) \text{ for any } x \in X,$$
$$(\bigvee_{i \in I} \mu_t)(x) = \bigvee_{t \in T} \mu_t(x), (\bigwedge_{t \in T} \mu_t)(x) = \bigwedge_{t \in T} \mu_t(x),$$
$$(\mu * \nu)(x) = \mu(x) * \nu(x), (\mu \to \nu)(x) = \mu(x) \to \nu(x).$$

We don't distinguish between a constant function and its value because no confusion will occur.

Let $f : X \longrightarrow Y$ be a function. We define $f^{\rightarrow} : L^X \longrightarrow L^Y$ and $f^{\leftarrow} : L^Y \longrightarrow L^X$ [35] by $f^{\rightarrow}(\mu)(y) = \bigvee_{f(x)=y} \mu(x)$ for $\mu \in L^X$ and $y \in Y$, and $f^{\leftarrow}(\nu)(x) = \nu(f(x))$ for $\nu \in L^Y$ and $x \in X$.

Let $\mu, \nu$ be $L$-fuzzy sets in $X$. The subsethood degree [37–40] of $\mu, \nu$, denoted by $S_X(\mu, \nu)$, is defined by

$$S_X(\mu, \nu) = \bigwedge_{x \in X} (\mu(x) \rightarrow \nu(x)).$$

### 2.1. ⊤-Filters and Stratified L-Filters

A filter on a set $X$ is an upper set of $(2^X, \subseteq)$ ($2^X$ denotes the power set of $X$) wich is closed for finite meets and does not contain the empty set. The conception of filter has been generalized to the fuzzy setting in two methods; prefilters (or ⊤-filters more general) and $L$-filters. Both prefilters (⊤-filters) and $L$-filters play important roles in the theory of fuzzy topology, see [26,27,34,35,41–44].

**Definition 1** ([35]). *A nonempty subset $\mathbb{F} \subseteq L^X$ is called a ⊤-filter on the set $X$ whenever:*
*(TF1) $\bigvee_{x \in X} \lambda(x) = \top$ for all $\lambda \in \mathbb{F}$, (TF2) $\lambda \wedge \mu \in \mathbb{F}$ for all $\lambda, \mu \in \mathbb{F}$,*
*(TF3) if $\lambda \in L^X$ such that $\bigvee_{\mu \in \mathbb{F}} S_X(\mu, \lambda) = \top$, then $\lambda \in \mathbb{F}$.*
*The set of all ⊤-filters on $X$ is denoted as $\mathbb{F}_L^{\top}(X)$.*

**Definition 2** ([35]). *A nonempty subset $\mathbb{B} \subseteq L^X$ is called a ⊤-filter base on the set $X$ provided:*
*(TB1) $\bigvee_{x \in X} \lambda(x) = \top$ for all $\lambda \in \mathbb{B}$, (TB2) if $\lambda, \mu \in \mathbb{B}$, then $\bigvee_{\nu \in \mathbb{B}} S_X(\nu, \lambda \wedge \mu) = \top$.*

Each ⊤-filter base generates a ⊤-filter $\mathbb{F}_{\mathbb{B}}$ defined by

$$\mathbb{F}_{\mathbb{B}} := \{\lambda \in L^X | \bigvee_{\mu \in \mathbb{B}} S_X(\mu, \lambda) = \top\}.$$

**Example 1** ([26,45]). *Let $f : X \longrightarrow Y$ be a function, $\mathbb{F} \in \mathbb{F}_L^{\top}(X)$ and $\mathbb{G} \in \mathbb{F}_L^{\top}(Y)$. Then*

*(1) The family $\{f^{\rightarrow}(\lambda) | \lambda \in \mathbb{F}\}$ forms a ⊤-filter base on $Y$, and the ⊤-filter $f^{\Rightarrow}(\mathbb{F})$ generated by it is called the image of $\mathbb{F}$ under $f$. It is easily seen that $\mu \in f^{\Rightarrow}(\mathbb{F}) \Longleftrightarrow f^{\leftarrow}(\mu) \in \mathbb{F}$.*

*(2) The family $\{f^{\leftarrow}(\mu) | \mu \in \mathbb{G}\}$ forms a ⊤-filter base on $Y$ if and only if $\bigvee_{y \in f(X)} \mu(y) = \top$ holds for all $\mu \in \mathbb{G}$, and the ⊤-filter $f^{\Leftarrow}(\mathbb{G})$ (if exists) generated by it is called the inverse image of $\mathbb{G}$ under $f$. Additionally, $\mathbb{G} \subseteq f^{\Rightarrow}(f^{\Leftarrow}(\mathbb{G}))$ holds whenever $f^{\Leftarrow}(\mathbb{G})$ exists. Particularly, $f^{\Leftarrow}(\mathbb{G})$ always exists and $\mathbb{G} = f^{\Rightarrow}(f^{\Leftarrow}(\mathbb{G}))$ if $f$ is surjective.*

*(3) For any $x \in X$, the family $[x]_{\top} =: \{\lambda \in L^X | \lambda(x) = \top\}$ is a ⊤-filter on $X$, and $f^{\Rightarrow}([x]_{\top}) = [f(x)]_{\top}$.*

A stratified $L$-filter [35] on a set $X$ is a function $\mathcal{F} : L^X \longrightarrow L$ such that: $\forall \lambda, \mu \in L^X, \forall \alpha \in L$, (LF1) $\mathcal{F}(\bot) = \bot, \mathcal{F}(\top) = \top$; (LF2) $\mathcal{F}(\lambda) \wedge \mathcal{F}(\mu) = \mathcal{F}(\lambda \wedge \mu)$; (LFs) $\mathcal{F}(\alpha * \lambda) \geq \alpha * \mathcal{F}(\lambda)$.

The set of all stratified $L$-filters on $X$ is denoted as $\mathcal{F}_L^s(X)$. A stratified $L$-filter $\mathcal{F}$ is called tight if $\mathcal{F}(\alpha) = \alpha$ for each $\alpha \in L$.

**Example 2** ([35]). *Let $f : X \longrightarrow Y$ be a function, $\mathcal{F} \in \mathcal{F}_L^s(X)$ and $\mathcal{G} \in \mathcal{F}_L^s(Y)$. Then*

*(1) The function $f^{\Rightarrow}(\mathcal{F}) : L^Y \longrightarrow L$ defined by $\mu \mapsto \mathcal{F}(\mu \circ f)$ is a stratified L-filter on $Y$ called the image of $\mathcal{F}$ under $f$.*

*(2) For any $x \in X$, the function $[x] : L^X \longrightarrow L, [x](\lambda) = \lambda(x)$ is a stratified L-filter on X, and $f^{\Rightarrow}([x]) = [f(x)]$.*

For each $\mathbb{F} \in \mathbb{F}_L^\top(X)$, define $\Lambda(\mathbb{F}) : L^X \longrightarrow L$ as

$$\forall \lambda \in L^X, \Lambda(\mathbb{F})(\lambda) = \bigvee_{\mu \in \mathbb{F}} S_X(\mu, \lambda),$$

then $\Lambda(\mathbb{F})$ is a tightly stratified $L$-filter on $X$ [44].

Conversely, for each tightly stratified $L$-filter $\mathcal{F}$ on a set $X$, the family

$$\Gamma(\mathcal{F}) = \{\lambda \in L^X, \mathcal{F}(\lambda) = \top\}$$

is a $\top$-filter on $X$ [44]. Given $\mathbb{F} \in \mathbb{F}_L^\top(X)$, we have $\Gamma\Lambda(\mathbb{F}) = \mathbb{F}$.

**Lemma 1.** *Let* $\{\mathbb{F}_j\}_{j \in J} \subseteq \mathbb{F}_L^\top(X)$. *If $L$ is continuous then* $\Lambda(\bigcap_{j \in J} \mathbb{F}_j) = \bigwedge_{j \in J} \Lambda(\mathbb{F}_j)$.

**Proof.** For any $\lambda \in L^X$ and any $j \in J$, note that $\{S_X(\mu_j, \lambda) | \mu_j \in \mathbb{F}_j\}$ is a directed subset of $L$. Then

$$
\begin{aligned}
\bigwedge_{j \in J} \Lambda(\mathbb{F}_j)(\lambda) &= \bigwedge_{j \in J} \bigvee_{\mu_j \in \mathbb{F}_j} S_X(\mu_j, \lambda) \overset{(\mathbf{DD})}{=} \bigvee_{h \in N} \bigwedge_{j \in J} S_X(h(j), \lambda) \\
&= \bigvee_{h \in N} S_X(\bigvee_{j \in J} h(j), \lambda), \text{ by } \bigvee_{j \in J} h(j) \in \bigcap_{j \in J} \mathbb{F}_j \\
&\leq \bigvee_{\nu \in \bigcap_{j \in J} \mathbb{F}_j} S_X(\nu, \lambda) \leq \Lambda(\bigcap_{j \in J} \mathbb{F}_j)(\lambda).
\end{aligned}
$$

Thus $\Lambda(\bigcap_{j \in J} \mathbb{F}_j)(\lambda) = \bigwedge_{j \in J} \Lambda(\mathbb{F}_j)(\lambda)$ since $\Lambda(\bigcap_{j \in J} \mathbb{F}_j)(\lambda) \leq \bigwedge_{j \in J} \Lambda(\mathbb{F}_j)(\lambda)$ holds obviously. □

### 2.2. $\top$-Convergence Spaces and Stratified L-Generalized Convergence Spaces

**Definition 3.** *A $\top$-convergence structure* [26] *on a set $X$ is a function* $q : \mathbb{F}_L^\top(X) \longrightarrow 2^X$ *satisfying*

**(TC1)** $[x]_\top \xrightarrow{q} x$ *for every* $x \in X$; **(TC2)** *if* $\mathbb{F} \xrightarrow{q} x$ *and* $\mathbb{F} \subseteq \mathbb{G}$, *then* $\mathbb{G} \xrightarrow{q} x$, *where* $\mathbb{F} \xrightarrow{q} x$ *is shorthand for* $x \in q(\mathbb{F})$. *The pair* $(X, q)$ *is called a $\top$-convergence space.*

A function $f : X \longrightarrow X'$ between two $\top$-convergence spaces $(X, q)$, $(X', q')$ is called continuous if $f^\Rightarrow(\mathbb{F}) \xrightarrow{q'} f(x)$ whenever $\mathbb{F} \xrightarrow{q} x$.

The category whose objects are $\top$-convergence spaces and whose morphisms are continuous functions will be denoted by $\top$-**CS**. This category is topological over **SET** [26,46].

For a given source $(X \xrightarrow{f_i} (X_i, q_i))_{i \in I}$, the initial structure [47], $q$ on $X$ is defined by

$$\mathbb{F} \xrightarrow{q} x \Longleftrightarrow \forall i \in I, f_i^\Rightarrow(\mathbb{F}) \xrightarrow{q_i} f_i(x).$$

For a given sink $((X_i, q_i) \xrightarrow{f_i} X)_{i \in I}$, the final structure, $q$ on $X$ is defined as

$$\mathbb{F} \xrightarrow{q} x \Longleftrightarrow \begin{cases} \mathbb{F} \supseteq [x]_\top, & x \notin \cup_{i \in I} f_i(X_i); \\ \mathbb{F} \supseteq f_i^\Rightarrow(\mathbb{G}_i), & \exists i \in I, x_i \in X_i, \mathbb{G}_i \in \mathbb{F}_L^\top(X_i) \text{ s.t } f(x_i) = x, \mathbb{G}_i \xrightarrow{q_i} x_i. \end{cases}$$

Thus, when $X = \cup_{i \in I} f_i(X_i)$, the final structure $q$ can be simplified as

$$\mathbb{F} \xrightarrow{q} x \Longleftrightarrow \mathbb{F} \supseteq f_i^\Rightarrow(\mathbb{G}_i) \text{ for some } \mathbb{G}_i \xrightarrow{q_i} x_i \text{ with } f(x_i) = x.$$

For a nonempty set $X$, we use $\top(X)$ to denote all $\top$-convergence structures on $X$. For $p, q \in \top(X)$, we say that $q$ is finer than $p$, or $p$ is coarser than $q$, denoted by $p \leq q$ for short, if the identity $\mathrm{id}_X : (X, q) \longrightarrow (X, p)$ is continuous, that is, $\mathbb{F} \xrightarrow{q} x \Longrightarrow \mathbb{F} \xrightarrow{p} x$. It is easily observed from [26,47] that $(\top(X), \leq)$ forms a completed lattice, and the discrete (resp., indiscrete) structure $\delta$ (resp., $\iota$) is the top (resp., bottom) element of $(T(X), \leq)$, where $\delta$ is given by $\mathbb{F} \xrightarrow{\delta} x$ iff $\mathbb{F} \supseteq [x]_\top$; and $\iota$ is given by $\mathbb{F} \xrightarrow{\iota} x$ for all $\mathbb{F} \in \mathbb{F}_L^\top(X)$, $x \in X$.

**Definition 4.** *(Jäger [5] and Yao [25]) A stratified L-generalized convergence structure on a set $X$ is a function* $\lim^q : \mathcal{F}_L^s(X) \longrightarrow L^X$ *satisfying*

**(LC1)** $\lim^q [x](x) = 1$ *for every $x \in X$;* **(LC2)** $\forall \mathcal{F}, \mathcal{G} \in \mathcal{F}_L^s(X), \mathcal{F} \leq \mathcal{G} \Longrightarrow \lim^q \mathcal{F} \leq \lim^q \mathcal{G}$.
*The pair $(X, \lim^q)$ is called a stratified L-generalized convergence space.*

Let $(X, q)$ be a $\top$-convergence space. We define $\lim^q : \mathcal{F}_L^s(X) \longrightarrow L^X$ as

$$\lim^q \mathcal{F}(x) = \begin{cases} \top, & \mathcal{F} \geq \Lambda(\mathbb{F}) \text{ for some } \mathbb{F} \xrightarrow{q} x; \\ \bot, & \text{otherwise.} \end{cases}$$

Note that $[x] = \Lambda([x]_\top)$. It follows that $(X, \lim^q)$ is a stratified $L$-generalized convergence space.

**Remark 1.** *When $L = \{\bot, \top\}$, both $\top$-convergence spaces and stratified L-generalized convergence spaces all reduce to convergence spaces. Therefore, these two kinds of lattice-valued convergence spaces are all natural extensions of convergence spaces.*

## 3. $p$-Topologicalness in $\top$-Convergence Spaces

In this section, we shall discuss the $p$-topologicalness in $\top$-convergence spaces by generalized Fischer's diagonal condition and generalized Gähler's neighborhood condition, respectively. We also try to establish the relationships between $p$-topologicalness in $\top$-convergence spaces and $p$-topologicalness in stratified $L$-generalized convergence spaces.

### 3.1. $p$-Pretopologicalness in $\top$-Convergence Spaces

Let $(X, p)$ be a $\top$-convergence space. Then for any $x \in X$, the $\top$-filter

$$\mathbb{U}_p(x) = \cap \{\mathbb{F} \in \mathbb{F}_L^\top(X) | \mathbb{F} \xrightarrow{p} x\}$$

is called the $\top$-neighborhood with respect to $p$ at $x$. Then the family $\mathbb{U}_p := \{\mathbb{U}_p(x)\}_{x \in X}$ is called the $\top$-neighborhood system generated by $(X, p)$ [26]. It is easily seen that if $p, p' \in T(X)$ and $p \leq p'$ then $\mathbb{U}_p(x) \subseteq \mathbb{U}_{p'}(x)$ for any $x \in X$.

In the following, we shorten a pair of $\top$-convergence spaces $(X, p)$ and $(X, q)$ as $(X, p, q)$. It is easy to check that the following conditions are equivalent:

$p$-**(TP1)**: $\forall \{\mathbb{F}_j\}_{j \in J} \subseteq \mathbb{F}_L^\top(X), \forall x \in X, \forall j \in J, \mathbb{F}_j \xrightarrow{p} x \Longrightarrow \cap_{j \in J} \mathbb{F}_j \xrightarrow{q} x$.
$p$-**(TP2)**: $\forall \mathbb{F} \in \mathbb{F}_L^\top(X), \forall x \in X, \mathbb{F} \supseteq \mathbb{U}_p(x) \Longrightarrow \mathbb{F} \xrightarrow{q} x$.
$p$-**(TP3)**: $\forall x \in X, \mathbb{U}_p(x) \xrightarrow{q} x$.

**Definition 5.** *Assume that $(X, p, q)$ is a pair of $\top$-convergence spaces. Then $q$ is said to be $p$-pretopological if it fulfills either of the above three conditions.*

**Remark 2.** *When $p = q$, p-pretopologicalness is precise the pretopologicalness in [26]. In this case, it is observed easily that the "$\Longrightarrow$" in p-**(TP2)** can be replaced with "$\Longleftrightarrow$". In the following, when $p = q$, we omit the prefix "p" in symbols p-**(TP1)**–p-**(TP3)**. This simplification is also used for the subsequent p-topological conditions.*

**Proposition 1.** *A $\top$-convergence structure $q$ on $X$ is pretopological iff it is p-pretopological for any $q \in \top(X)$ with $q \leq p$.*

**Proof.** Let $(X, q)$ be pretopological and $q \leq p$. Then by $q \leq p$ we have $\mathbb{U}_p(x) \supseteq \mathbb{U}_q(x)$ for any $x \in X$. By pretopologicalness of $q$ we get that $\mathbb{U}_q(x) \xrightarrow{q} x$. It follows that $\mathbb{U}_p(x) \xrightarrow{q} x$. Thus $q$ is $p$-pretopological. The converse implication is obvious. $\square$

The following example shows there is no $p$-pretopologicalness implies pretopologicalness in general.

**Example 3.** *Let L be the linearly ordered frame $(\{\perp, \alpha, \top\}, \wedge, \top)$ with $\perp < \alpha < \top$, and $X = \{x, y\}$. For each $\mathbb{F} \in \mathbb{F}_L^\top(X)$ and $z \in X$, let $\mathbb{F} \xrightarrow{p} z \Longleftrightarrow \mathbb{F} \supseteq [z]$. In [26], it is proved that $(X, p)$ is a $\top$-convergence space and for each $z \in X, \mathbb{U}_p(z) = [z]$.*

*For $x, y \in X$, it is easily seen that the subsets $\mathbb{F}_x, \mathbb{F}_y$ of $L^X$ defined by*

$$\mathbb{F}_x = \{\lambda \in L^X : \lambda(x) \geq \alpha, \lambda(y) = \top\}; \mathbb{F}_y(\lambda) = \{\lambda \in L^X : \lambda(y) \geq \alpha, \lambda(x) = \top\}$$

*are all $\top$-filters on $X$. For each $\mathbb{F} \in \mathbb{F}_L^\top(X)$ and each $z \in X$, let $\mathbb{F} \xrightarrow{q} z \Longleftrightarrow \mathbb{F} \supseteq [z]$ or $\mathbb{F} \supseteq \mathbb{F}_z$. Then $(X, q)$ is a $\top$-convergence space. For each $z \in X$, $\mathbb{U}_q(z) = [z] \cap \mathbb{F}_z = \{\top_X\}$ and so $[z] \cap \mathbb{F}_z \neq [z], \mathbb{F}_z$.*

*Obviously, $q$ satisfies $p$-**(TP3)**. But $q$ is not pretopological since we have no $\mathbb{U}_q(z) \xrightarrow{q} z$.*

### 3.2. p-Topologicalness in $\top$-Convergence Spaces

At first, we fix the notions of diagonal $\top$-filter and neighborhood $\top$-filter to state $p$-topologicalness. Let $J, X$ be any sets and $\phi : J \longrightarrow \mathbb{F}_L^\top(X)$ be any function. Then a function $\hat{\phi} : L^X \to L^J$ is defined as

$$\forall \lambda \in L^X, \forall j \in J, \hat{\phi}(\lambda)(j) = \Lambda(\phi(j))(\lambda) = \bigvee_{\mu \in \phi(j)} S_X(\mu, \lambda).$$

For all $\mathbb{F} \in \mathbb{F}_L^\top(J)$, it is proved that a subset of $L^X$ defined by

$$k\phi\mathbb{F} := \{\lambda \in L^X | \hat{\phi}(\lambda) \in \mathbb{F}\}$$

is a $\top$-filter, called diagonal $\top$-filter of $\mathbb{F}$ under $\phi$ [26]. In addition, for any $\lambda, \mu \in L^X$, it was proved in [26] that $S_X(\lambda, \mu) \leq S_J(\hat{\phi}(\lambda), \hat{\phi}(\mu))$.

**Definition 6** ([34])**.** *Let $(X, p)$ be a $\top$-convergence space and $\mathbb{U}_p : X \longrightarrow \mathbb{F}_L^\top(X)$ be the $\top$-neighborhood system generated by $(X, p)$. Then for each $\mathbb{F} \in \mathbb{F}_L^\top(X)$, the $\top$-filter $\mathbb{U}_p(\mathbb{F}) := k\mathbb{U}_p\mathbb{F}$, is called neighborhood $\top$-filter of $\mathbb{F}$ w.r.t. p.*

Let $\mathbb{N}$ be the set of natural numbers including 0. Let $(X, p)$ be a $\top$-convergence space and $\mathbb{F} \in \mathbb{F}_L^\top(X)$. For any $n \in \mathbb{N}$, we define $\mathbb{U}_p^0(\mathbb{F}) = \mathbb{F}$, and if $\mathbb{U}_p^n(\mathbb{F})$ has been defined, then we define the $n + 1$ th iteration of the neighborhood $\top$-filter of $\mathbb{F}$ inductive by $\mathbb{U}_p^{n+1}(\mathbb{F}) = \mathbb{U}_p(\mathbb{U}_p^n(\mathbb{F}))$.

**Proposition 2.** *Let $(X, p)$ be a $\top$- convergence space, $n \in \mathbb{N}$ and $\mathbb{F}, \mathbb{G} \in \mathbb{F}_L^\top(X)$. Then*
    (1) $\mathbb{U}_p^n(\mathbb{F}) \subseteq \mathbb{F}$,

(2) *if* $\mathbb{F} \subseteq \mathbb{G}$, *then* $\mathbb{U}_p^n(\mathbb{F}) \subseteq \mathbb{U}_p^n(\mathbb{G})$,

(3) *if* $p' \in T(X)$ *and* $p \leq p'$, *then* $\mathbb{U}_p^n(\mathbb{F}) \subseteq \mathbb{U}_{p'}^n(\mathbb{F})$.

**Proof.** It is obvious. $\square$

**Definition 7.** *Let* $f : (X, q) \longrightarrow (Y, p)$ *be a function between* $\top$-*convergence spaces. Then* $f$ *is said to be an interior function if* $f^\rightarrow(\widehat{\mathbb{U}}_q(\lambda)) \leq \widehat{\mathbb{U}}_p(f^\rightarrow(\lambda))$ *for all* $\lambda \in L^X$.

**Proposition 3.** *Let* $f : (X, q) \longrightarrow (Y, p)$ *be a function between* $\top$-*convergence spaces and* $\mathbb{F} \in \mathbb{F}_L^\top(X)$.

(1) *If* $f$ *is continuous, then* $f^\Rightarrow(\mathbb{U}_q^n(\mathbb{F})) \supseteq \mathbb{U}_p^n(f^\Rightarrow(\mathbb{F}))$.

(2) *If* $f$ *is an interior function, then* $f^\Rightarrow(\mathbb{U}_q^n(\mathbb{F})) \subseteq \mathbb{U}_p^n(f^\Rightarrow(\mathbb{F}))$.

**Proof.** (1) We prove $f^\Rightarrow(\mathbb{U}_q^n(\mathbb{F})) \supseteq \mathbb{U}_p^n(f^\Rightarrow(\mathbb{F}))$ inductively.

For each $\mathbb{F} \xrightarrow{q} x$ and each $\lambda \in \mathbb{U}_p(f(x))$ we have $\lambda \in f^\Rightarrow(\mathbb{F})$, i.e., $f^\leftarrow(\lambda) \in \mathbb{F}$ and then

$$f^\leftarrow(\lambda) \in \cap\{\mathbb{F}|\mathbb{F} \xrightarrow{q} x\} = \mathbb{U}_q(x),$$

i.e., $\lambda \in f^\Rightarrow(\mathbb{U}_q(x))$. Thus $\mathbb{U}_p(f(x)) \subseteq f^\Rightarrow(\mathbb{U}_q(x))$.

Fixing $\lambda \in L^Y$, we get

$$\widehat{\mathbb{U}_p}(\lambda)(f(x)) = \bigvee_{\mu \in \mathbb{U}_p(f(x))} S_Y(\mu, \lambda) \leq \bigvee_{f^\leftarrow(\mu) \in \mathbb{U}_q(x)} S_X(f^\leftarrow(\mu), f^\leftarrow(\lambda)) \leq$$
$$\bigvee_{\nu \in \mathbb{U}_q(x)} S_X(\nu, f^\leftarrow(\lambda)) = \widehat{\mathbb{U}_q}(f^\leftarrow(\lambda))(x).$$

It follows that $f^\leftarrow(\widehat{\mathbb{U}_p}(\lambda)) = \widehat{\mathbb{U}_q}(f^\leftarrow(\lambda))$. Thus

$$\lambda \in \mathbb{U}_p(f^\Rightarrow(\mathbb{F})) \implies \widehat{\mathbb{U}_p}(\lambda) \in f^\Rightarrow(\mathbb{F}) \implies f^\leftarrow(\widehat{\mathbb{U}_p}(\lambda)) \in \mathbb{F} \implies \widehat{\mathbb{U}_q}(f^\leftarrow(\lambda)) \in \mathbb{F}$$
$$\implies f^\leftarrow(\lambda) \in \mathbb{U}_q(\mathbb{F}) \implies \lambda \in f^\Rightarrow(\mathbb{U}_q(\mathbb{F})).$$

So, $f^\Rightarrow(\mathbb{U}_q^n(\mathbb{F})) \supseteq \mathbb{U}_p^n(f^\Rightarrow(\mathbb{F}))$ when $n = 1$.

We assume that $f^\Rightarrow(\mathbb{U}_q^n(\mathbb{F})) \supseteq \mathbb{U}_p^n(f^\Rightarrow(\mathbb{F}))$ when $n = k$. Then we need to check that $f^\Rightarrow(\mathbb{U}_q^n(\mathbb{F})) \supseteq \mathbb{U}_p^n(f^\Rightarrow(\mathbb{F}))$ when $n = k + 1$. Indeed,

$$f^\Rightarrow(\mathbb{U}_q^{k+1}(\mathbb{F})) = f^\Rightarrow(\mathbb{U}_q(\mathbb{U}_q^k(\mathbb{F}))) \supseteq \mathbb{U}_p(f^\Rightarrow(\mathbb{U}_q^k(\mathbb{F}))) \supseteq \mathbb{U}_p(\mathbb{U}_p^k(f^\Rightarrow(\mathbb{F}))) = \mathbb{U}_p^{k+1}(f^\Rightarrow(\mathbb{F})).$$

(2) We check only the inequalities for $n = 1$.

Let $f$ be an interior function. For each $\lambda \in \mathbb{U}_q(\mathbb{F})$, i.e., $\widehat{\mathbb{U}_q}(\lambda) \in \mathbb{F}$ we have $f^\rightarrow(\widehat{\mathbb{U}_q}(\lambda)) \in f^\Rightarrow(\mathbb{F})$ and then $\widehat{\mathbb{U}_p}(f^\rightarrow(\lambda)) \in f^\Rightarrow(\mathbb{F})$ by $f$ is an interior function. That means $f^\leftarrow(\lambda) \in \mathbb{U}_p(f^\Rightarrow(\mathbb{F}))$. Thus $f^\Rightarrow(\mathbb{U}_q(\mathbb{F})) \subseteq \mathbb{U}_p(f^\Rightarrow(\mathbb{F}))$. $\square$

Now, we tend our attention to $p$-topologicalness.

We say a pair of $\top$-convergence spaces $(X, p, q)$ satisfy the Gähler $\top$-neighborhood condition if

$p$-**(TG)**: $\forall \mathbb{F} \in \mathbb{F}_L^\top(X), \forall x \in X, \mathbb{F} \xrightarrow{q} x \implies \mathbb{U}_p(\mathbb{F}) \xrightarrow{q} x$.

**Definition 8.** *Let* $(X, p, q)$ *be a pair of* $\top$-*convergence spaces. Then* $q$ *is called* $p$-*topological if the condition* $p$-*(TG) is satisfied.*

**Remark 3.** *When $L = \{\bot, \top\}$, the condition p-(TG) is precise the Gähler neighborhood condition in [4], which is used to define p-topological convergence spaces. Therefore, our p-topologicalness is a natural extension of crisp p-topologicalness.*

We say a pair of $\top$-convergence spaces $(X, p, q)$ satisfy the Fischer $\top$-diagonal condition if

$p$**-(TF)**: Let $J, X$ be any sets, $\psi : J \longrightarrow X$, and $\phi : J \longrightarrow \mathbb{F}_L^\top(X)$ such that $\phi(j) \xrightarrow{p} \psi(j)$, for each $j \in J$. Then for each $\mathbb{F} \in \mathbb{F}_L^\top(J)$ and each $x \in X$, $\psi^\Rightarrow(\mathbb{F}) \xrightarrow{q} x$ implies $k\phi\mathbb{F} \xrightarrow{q} x$.

Restricting $J = X$ and $\psi =$ id in $p$-**(TF)**, we obtain a weaker condition $p$-**(TK)**. When $p = q$, $p$-**(TF)** is precise the Fischer $\top$-diagonal condition **(TF)**, and $p$-**(TK)** is precise the Kowalsky $\top$-diagonal condition **(TK)** in [26].

**Proposition 4.** *Let $(X, p, q)$ be a pair of $\top$-convergence spaces. Then (1) p-(TF)$\Longrightarrow$p-(TP1)+p-(TK), and (2) p-(TK)$\Longrightarrow$p-(TF) if p satisfies (TP1).*

**Proof.** (1) Obviously, $p$-(TF)$\Longrightarrow p$-(TK). Now, we check $p$-(TF)$\Longrightarrow p$-(TP1). Let $\{\mathbb{F}_j\}_{j \in J} \subseteq \mathbb{F}_L^\top(X)$ and $x \in X$ satisfy $\forall j \in J$, $\mathbb{F}_j \xrightarrow{p} x$. Take $\psi(j) \equiv x$, $\phi(j) = \mathbb{F}_j$ and $\mathbb{F} = \mathbb{F}_\bot$ (i.e., $\mathbb{F}_\bot = \{\top_J\}$, the smallest $\top$-filter on $J$) in $p$-(TF), then it is easily seen that $\psi^\Rightarrow(\mathbb{F}_\bot) = [x]_\top$ and $k\phi\mathbb{F}_\bot = \bigcap_{j \in J}\mathbb{F}_j$. Because $\psi^\Rightarrow(\mathbb{F}_\bot) = [x]_\top \xrightarrow{q} x$ we have $k\phi\mathbb{F}_\bot = \bigcap_{j \in J}\mathbb{F}_j \xrightarrow{q} x$ by $p$-(TF).

(2) Let $J, X, \psi, \phi$ satisfy the condition of $p$-(TF). Then we define a function $\widetilde{\phi} : X \longrightarrow \mathbb{F}_L^\top(X)$ as $\widetilde{\phi}(x) = \bigcap\{\phi(j) : j \in J, \psi(j) = x\}$ if there exists $j \in J$ such that $\psi(j) = x$ and $\widetilde{\phi}(x) = [x]_\top$ if not so. For each $x \in X$, if $\widetilde{\phi}(x) = [x]_\top$ then $\widetilde{\phi}(x) \xrightarrow{p} x$. If $\widetilde{\phi}(x) = \bigcap\{\phi(j) : j \in J, \psi(j) = x\}$ then by $\phi(j) \xrightarrow{p} x$ and (TP1) we have $\widetilde{\phi}(x) \xrightarrow{p} x$. Let $\mathbb{F} \in \mathbb{F}_L^\top(J)$ and $\psi^\Rightarrow(\mathbb{F}) \xrightarrow{q} x$. Then by $p$-(TK) we obtain $k\widetilde{\phi}\psi^\Rightarrow(\mathbb{F}) \xrightarrow{q} x$. One can prove that $k\phi\mathbb{F} \supseteq k\widetilde{\phi}\psi^\Rightarrow(\mathbb{F})$. Thus $k\phi\mathbb{F} \xrightarrow{q} x$. $\square$

**Corollary 1.** *Let $(X, p, q)$ be a pair of $\top$-convergence spaces. If p satisfies (TP1) then p-(TF)$\Leftrightarrow$p-(TK)+p-(TP1). In particular, when $p = q$ we have (TF)$\Longleftrightarrow$(TK)+(TP1) [26].*

**Remark 4.** *Let L, X and $\mathbb{F}_z(z \in X)$ be defined as in Example 3. Let q be defined as $\mathbb{F} \xrightarrow{q} z$ for any $\mathbb{F} \in \mathbb{F}_L^\top(X)$ and any $z \in X$, and let p be defined as $\mathbb{F} \xrightarrow{p} z \Longleftrightarrow \mathbb{F} \supset \mathbb{F}_\bot$. Then $(X, p, q)$ is a pair of $\top$-convergence spaces. Obviously, the axiom p-(TF) is satisfied. But p does not fulfill the axiom (TP1) since $\mathbb{U}_p(z) = \mathbb{F}_\bot \xrightarrow{p}\!\!\!\!\!\!/\;\; z$. Thus this example shows that p-(TF) does not imply (TP1) of $(X, p)$ generally. Therefore, we guess that the additional condition (TP1) in the above corollary can not be removed.*

The following theorem shows that if we restricting the lattice-context slightly, $p$-topologicalness can be described by Fischer $\top$-diagonal condition $p$-**(TF)**.

**Theorem 1.** *Let $(X, p, q)$ be a pair of $\top$-convergence spaces. Then p-(TG)$\Longrightarrow$p-(TF), and the converse inclusion holds if L is continuous.*

**Proof.** *p*-**(TG)**$\Longrightarrow$*p*-**(TF)**. Let $J, X, \phi, \psi$ satisfy the condition of *p*-**(TF)**. For any $\mathbb{F} \in \mathbb{F}_L^\top(J)$, we prove below that $\mathbb{U}_p(\psi^\Rightarrow(\mathbb{F})) \subseteq k\phi\mathbb{F}$. Let $\lambda \in L^X$,

$$
\begin{aligned}
\bigvee_{\mu \in \mathbb{F}} S_X(\psi^\rightarrow(\mu), \widehat{\mathbb{U}_p}(\lambda)) &= \bigvee_{\mu \in \mathbb{F}} \bigwedge_{x \in X} ((\bigvee_{\psi(j)=x} \mu(j)) \rightarrow \bigvee_{\nu \in \mathbb{U}_p(x)} S_X(\nu, \lambda)) \\
&= \bigvee_{\mu \in \mathbb{F}} \bigwedge_{j \in J} (\mu(j) \rightarrow \bigvee_{\nu \in \mathbb{U}_p(\psi(j))} S_X(\nu, \lambda)), \text{ by } \phi(j) \xrightarrow{p} \psi(j) \\
&\leq \bigvee_{\mu \in \mathbb{F}} \bigwedge_{j \in J} (\mu(j) \rightarrow \bigvee_{\nu \in \phi(j)} S_X(\nu, \lambda)) \\
&= \bigvee_{\mu \in \mathbb{F}} \bigwedge_{j \in J} (\mu(j) \rightarrow \hat{\phi}(\lambda)(j)) = \bigvee_{\mu \in \mathbb{F}} S_J(\mu, \hat{\phi}(\lambda)).
\end{aligned}
$$

It follows that

$$
\lambda \in \mathbb{U}_p(\psi^\Rightarrow(\mathbb{F})) \implies \bigvee_{\mu \in \mathbb{F}} S_X(\psi^\rightarrow(\mu), \widehat{\mathbb{U}_p}(\lambda)) = \top \implies \bigvee_{\mu \in \mathbb{F}} S_J(\mu, \hat{\phi}(\lambda)) = \top \implies \lambda \in k\phi\mathbb{F}.
$$

Thus $\mathbb{U}_p(\psi^\Rightarrow(\mathbb{F})) \subseteq k\phi\mathbb{F}$.

If $\psi^\Rightarrow(\mathbb{F}) \xrightarrow{q} x$ then it follows by *p*-**(TG)** that $\mathbb{U}_p(\psi^\Rightarrow(\mathbb{F})) \xrightarrow{q} x$, and so $k\phi\mathbb{F} \xrightarrow{q} x$. That is, *p*-**(TF)** is satisfied.

*p*-**(TF)**$\Longrightarrow$*p*-**(TG)**. Note that Lemma 1 holds since $L$ is continuous. Take

$$
J = \{(\mathbb{G}, y) \in \mathbb{F}_L^\top(X) \times X | \mathbb{G} \xrightarrow{p} y\}; \psi : J \longrightarrow X, (\mathbb{G}, y) \mapsto y; \phi : J \longrightarrow \mathbb{F}_L^\top(X), (\mathbb{G}, y) \mapsto \mathbb{G}.
$$

Then $\forall j \in J$, $\phi(j) \xrightarrow{p} \psi(j)$. Because $[y] \xrightarrow{p} y$ we have that $\psi$ is a surjective function. Thus for each $\mathbb{F} \in \mathbb{F}_L^\top(X)$, $\mathbb{H} = \psi^\Leftarrow(\mathbb{F}) \in \mathbb{F}_L^\top(J)$ exists and $\psi^\Rightarrow(\mathbb{H}) = \mathbb{F}$.

We prove below that $k\phi\mathbb{H} = \mathbb{U}_p(\mathbb{F})$. For any $y \in X$, denote $I_y = \{\mathbb{G} \in \mathbb{F}_L^\top(X) | \mathbb{G} \xrightarrow{p} y\}$. Then for any $\lambda \in L^X$,

$$
\begin{aligned}
\bigvee_{\mu \in \mathbb{F}} S_J(\psi^\Leftarrow(\mu), \hat{\phi}(\lambda)) &= \bigvee_{\mu \in \mathbb{F}} \bigwedge_{(\mathbb{G}, y) \in J} (\psi^\Leftarrow(\mu)(\mathbb{G}, y) \rightarrow \hat{\phi}(\lambda)(\mathbb{G}, y)) \\
&= \bigvee_{\mu \in \mathbb{F}} \bigwedge_{y \in X} \bigwedge_{\mathbb{G} \in I_y} (\mu(y) \rightarrow \Lambda(\mathbb{G})(\lambda)) \\
&= \bigvee_{\mu \in \mathbb{F}} \bigwedge_{y \in X} (\mu(y) \rightarrow \bigwedge_{\mathbb{G} \in I_y} \Lambda(\mathbb{G})(\lambda)), \text{ by Lemma 1} \\
&= \bigvee_{\mu \in \mathbb{F}} \bigwedge_{y \in X} (\mu(y) \rightarrow \Lambda(\bigcap_{\mathbb{G} \in I_y} \mathbb{G})(\lambda)) \\
&= \bigvee_{\mu \in \mathbb{F}} \bigwedge_{y \in X} (\mu(y) \rightarrow \Lambda(\mathbb{U}_p(y))(\lambda)) \\
&= \bigvee_{\mu \in \mathbb{F}} \bigwedge_{y \in X} (\mu(y) \rightarrow \widehat{\mathbb{U}_p}(\lambda)(y)) = \bigvee_{\mu \in \mathbb{F}} S_X(\mu, \widehat{\mathbb{U}_p}(\lambda)).
\end{aligned}
$$

It follows that

$$
\begin{aligned}
\lambda \in k\phi\mathbb{H} &\iff \hat{\phi}(\lambda) \in \psi^\Leftarrow(\mathbb{F}) \iff \bigvee_{\mu \in \mathbb{F}} S_J(\psi^\Leftarrow(\mu), \hat{\phi}(\lambda)) = \top \iff \bigvee_{\mu \in \mathbb{F}} S_X(\mu, \widehat{\mathbb{U}_p}(\lambda)) = \top \\
&\iff \widehat{\mathbb{U}_p}(\lambda) \in \mathbb{F} \iff \lambda \in \mathbb{U}_p(\mathbb{F}).
\end{aligned}
$$

Thus $k\phi\mathbb{H} = \mathbb{U}_p(\mathbb{F})$.

Let $\mathbb{F} = \psi^{\Rightarrow}(\mathbb{H}) \xrightarrow{q} x$. Then by $p$-**(TF)** we have $k\phi\mathbb{H} = \mathbb{U}_p(\mathbb{F}) \xrightarrow{q} x$. That is, $p$-**(TG)** holds.  $\square$

The following theorem shows that for pretopological $\top$-convergence spaces, $p$-topologicalness can be described by Fischer $\top$-diagonal condition $p$-**(TF)**.

**Theorem 2.** *Let* $(X, p, q)$ *be a pair of $\top$-convergence spaces and* $(X, p)$ *be pretopological. Then* $p$-(TF)$\Longleftrightarrow$$p$-(TG).

**Proof.** Most of the proof can copy that of Theorem 1. We only check that

$$\bigvee_{\mu\in\mathbb{F}} S_J(\psi^{\leftarrow}(\mu), \hat{\phi}(\lambda)) = \bigvee_{\mu\in\mathbb{F}} S_X(\mu, \widehat{\mathbb{U}_p}(\lambda)),$$

for any $\lambda \in L^X$ in $p$-**(TF)**$\Longrightarrow$$p$-**(TG)**. Indeed, since $p$ is pretopological then $\mathbb{U}_p(y) \in I_y$ for any $y \in X$. Thus

$$
\begin{aligned}
\bigvee_{\mu\in\mathbb{F}} S_J(\psi^{\leftarrow}(\mu), \hat{\phi}(\lambda)) &= \bigvee_{\mu\in\mathbb{F}} \bigwedge_{(\mathbb{G},y)\in J} (\psi^{\leftarrow}(\mu)(\mathbb{G}, y) \to \hat{\phi}(\lambda)(\mathbb{G}, y)) \\
&= \bigvee_{\mu\in\mathbb{F}} \bigwedge_{y\in X} \bigwedge_{\mathbb{G}\in I_y} (\mu(y) \to \Lambda(\mathbb{G})(\lambda)) \\
&= \bigvee_{\mu\in\mathbb{F}} \bigwedge_{y\in X} (\mu(y) \to \bigwedge_{\mathbb{G}\in I_y} \Lambda(\mathbb{G})(\lambda)), \text{ by } \mathbb{U}_p(y) \in I_y \\
&= \bigvee_{\mu\in\mathbb{F}} \bigwedge_{y\in X} (\mu(y) \to \Lambda(\mathbb{U}_p(y))(\lambda)) \\
&= \bigvee_{\mu\in\mathbb{F}} S_X(\mu, \widehat{\mathbb{U}_p}(\lambda)).  \quad \square
\end{aligned}
$$

By Corollary 1 and Theorem 2 we get the following corollary.

**Corollary 2.** *[34] Let* $(X, p)$ *be a $\top$-convergence space. Then (TF)$\Longleftrightarrow$(TG).*

**Remark 5.** *The above corollary is one of the main results in [34]. Based on this equivalence, it was proved that $\top$-convergence spaces with (TF) or (TG) characterize precisely the conical L-topological spaces in [44].*

The following theorem shows that $p$-topologicalness is preserved under initial constructions.

**Theorem 3.** *Let* $\{(X_i, q_i, p_i)\}_{i\in I}$ *be pairs of $\top$-convergence spaces with each $q_i$ being $p_i$-topological. If $q$ (resp., $p$) is the initial structure on $X$ relative to the source* $(X \xrightarrow{f_i} (X_i, q_i))_{i\in I}$ *(resp.,* $(X \xrightarrow{f_i} (X_i, p_i))_{i\in I}$), *then* $(X, q)$ *is $p$-topological.*

**Proof.** Let $\mathbb{F} \xrightarrow{q} x$. Then by definition of $q$, we have $f_i^{\Rightarrow}(\mathbb{F}) \xrightarrow{q_i} f_i(x)$ for any $i \in I$. Because $q_i$ is $p_i$-topological we have $\mathbb{U}_{p_i}(f_i^{\Rightarrow}(\mathbb{F})) \xrightarrow{q_i} f_i(x)$. Then by Proposition 3 (1) we have $f_i^{\Rightarrow}(\mathbb{U}_p(\mathbb{F})) \supseteq \mathbb{U}_{p_i}(f_i^{\Rightarrow}(\mathbb{F}))$ and so $f_i^{\Rightarrow}(\mathbb{U}_p(\mathbb{F})) \xrightarrow{q_i} f_i(x)$ for all $i \in I$. That is, $\mathbb{U}_p(\mathbb{F}) \xrightarrow{q} x$. Thus $q$ is $p$-topological.  $\square$

The next theorem shows that $p$-topologicalness is preserved under final constructions with some additional conditions.

**Theorem 4.** *Let* $\{(X_i, q_i, p_i)\}_{i\in I}$ *be pairs of $\top$- convergence spaces with each $q_i$ being $p_i$-topological. Let $q$ (resp., $p$) be the final structure on $X$ w.r.t. The sink* $((X_i, q_i) \xrightarrow{f_i} X)_{i\in I}$ *(resp.,* $((X_i, p_i) \xrightarrow{f_i} X)_{i\in I}$). *If $X = \cup_{i\in I} f_i(X_i)$ and each $f_i : (X_i, p_i) \longrightarrow (X, p)$ is an interior function, then* $(X, q)$ *is $p$-topological.*

**Proof.** Let $\mathbb{F} \xrightarrow{q} x$. Then by definition of $q$, there exists $i \in I, x_i \in X_i, \mathbb{G}_i \in \mathbb{F}_L^\top(X_i)$ such that $f_i(x_i) = x, f_i^\Rightarrow(\mathbb{G}_i) \subseteq \mathbb{F}$ and $\mathbb{G}_i \xrightarrow{q_i} x_i$.

By $f_i^\Rightarrow(\mathbb{G}_i) \subseteq \mathbb{F}$ and $f_i$ is a interior function we have $f_i^\Rightarrow(\mathbb{U}_{p_i}(\mathbb{G}_i)) \subseteq \mathbb{U}_p(f_i^\Rightarrow(\mathbb{G}_i)) \subseteq \mathbb{U}_p(\mathbb{F})$.

By $\mathbb{G}_i \xrightarrow{q_i} x_i$ and $q_i$ is $p_i$-topological we have $\mathbb{U}_{p_i}(\mathbb{G}_i) \xrightarrow{q_i} x_i$, and then $f_i^\Rightarrow(\mathbb{U}_{p_i}(\mathbb{G}_i)) \xrightarrow{q} f_i(x_i) = x$.

Then it follows that $\mathbb{U}_p(\mathbb{F}) \xrightarrow{q} x$. By Theorem 1 we get that $q$ is $p$-topological. $\qquad\square$

From Theorem 3 and Theorem 4, we conclude easily the following corollary. It will tell us that $p$-topologicalness is preserved under supremum and infimum in the lattice $\top(X)$.

**Corollary 3.** *Let $\{q_i | i \in I\} \subseteq \top(X)$ and $p \in \top(X)$ such that each $(X, q_i)$ is $p$-topological. Then both $(X, \inf\{q_i\}_{i \in I})$ and $(X, \sup\{q_i\}_{i \in I})$ are all $p$-topological.*

*3.3. On the Relationship between p-Topologicalness in $\top$-Convergence Spaces and in Stratified L-Generalized Convergence Spaces*

Let $J, X$ be any set and $\Phi : J \longrightarrow \mathcal{F}_L^s(X)$ be any function. Then a function $\hat{\Phi} : L^X \to L^J$ is defined as $\forall \lambda \in L^X, \forall j \in J, \hat{\Phi}(\lambda)(j) = \Phi(j)(\lambda)$. For all $\mathcal{F} \in \mathcal{F}_L^s(J)$, it is proved that the function $K\Phi\mathcal{F} : L^X \longrightarrow L$ defined by $\forall \lambda \in L^X, K\Phi\mathcal{F}(\lambda) = \mathcal{F}(\hat{\Phi}(\lambda))$ is a stratified $L$-filter, which is called the diagonal $L$-filter of $\mathcal{F}$ under $\Phi$ [27,30].

Let $(X, \lim^p)$ be a stratified $L$-generalized convergence space. For any $\alpha \in L, x \in X$, let $\mathcal{U}_p^\alpha(x) = \bigwedge\{\mathcal{F} : \lim^p \mathcal{F}(x) \geq \alpha\}$. Take $\Phi = \mathcal{U}_p^\alpha : X \longrightarrow \mathcal{F}_L^s(X)$, then for each $\mathcal{F} \in \mathcal{F}_L^s(X)$, the stratified $L$-filter $\mathcal{U}_p^\alpha(\mathcal{F}) := k\mathcal{U}_p^\alpha\mathcal{F}$ is called $\alpha$-level neighborhood $L$-filter of $\mathcal{F}$ w.r.t. $\lim^p$ [29].

We say a pair of stratified $L$-generalized convergence spaces $(X, \lim^p, \lim^q)$ satisfy the Fischer $L$-diagonal condition if

$p$-**(LF)**: Let $J, X$ be any sets, $\Psi : J \longrightarrow X$ and $\Phi : J \longrightarrow \mathcal{F}_L^s(X)$ be functions.

$$\forall \mathcal{F} \in \mathcal{F}_L^s(J), \forall x \in X, \lim^q \Psi^\Rightarrow(\mathcal{F})(x) \wedge \bigwedge_{j \in J} \lim^p \Phi(j)(\Psi(j)) \leq \lim^q K\Phi\mathcal{F}(x).$$

We say a pair of stratified $L$-generalized convergence spaces $(X, \lim^p, \lim^q)$ satisfy the Gähler $L$-neighborhood condition if $p$-(LG): $\forall \alpha \in L, \forall \mathcal{F} \in \mathcal{F}_L^s(X), \alpha * \lim^q \mathcal{F} \leq \lim^q \mathcal{U}_p^\alpha(\mathcal{F})$.

It was proved in [32] that $p$-(LF) $\Longleftrightarrow$ $p$-(LG).

**Definition 9** ([32]). *Let $(X, \lim^p, \lim^q)$ be a pair of stratified L-generalized convergence spaces. Then $\lim^q$ is called $p$-topological if the condition $p$-(LF) or $p$-(LG) is satisfied.*

**Lemma 2.** *Let $\phi : J \longrightarrow \mathbb{F}_L^\top(X)$ be any function and $\mathbb{F} \in \mathbb{F}_L^\top(J)$. Then*
(1) $\Lambda(k\phi\mathbb{F}) \leq K(\Lambda \circ \phi)\Lambda(\mathbb{F})$;
(2) $k\phi\mathbb{F} = \Gamma(K(\Lambda \circ \phi)\Lambda(\mathbb{F}))$.

**Proof.** (1) Let $\lambda \in L^X$. Then for any $j \in J, \hat{\phi}(\lambda)(j) = \Lambda(\phi(j))(\lambda) = (\Lambda \circ \phi)(j)(\lambda) = \widehat{\Lambda \circ \phi}(\lambda)(j)$. It follows

$$
\begin{aligned}
\Lambda(k\phi\mathbb{F})(\lambda) &= \bigvee_{\mu \in k\phi\mathbb{F}} S_X(\mu, \lambda) = \bigvee_{\hat{\phi}(\mu) \in \mathbb{F}} S_X(\mu, \lambda) \\
&\leq \bigvee_{\hat{\phi}(\mu) \in \mathbb{F}} S_J(\hat{\phi}(\mu), \hat{\phi}(\lambda)) \leq \bigvee_{\nu \in \mathbb{F}} S_J(\nu, \hat{\phi}(\lambda)) \\
&= \bigvee_{\nu \in \mathbb{F}} S_J(\nu, \widehat{\Lambda \circ \phi}(\lambda)) = \Lambda(\mathbb{F})(\widehat{\Lambda \circ \phi}(\lambda)) = K(\Lambda \circ \phi)\Lambda(\mathbb{F})(\lambda).
\end{aligned}
$$

(2) Let $\lambda \in L^X$. Then

$$\lambda \in k\phi\mathbb{F} \iff \widehat{\Lambda \circ \phi}(\lambda) \in \mathbb{F} \iff \widehat{\Lambda \circ \phi}(\lambda) \in \mathbb{F} \iff \Lambda(\mathbb{F})(\widehat{\Lambda \circ \phi}(\lambda)) = \top$$
$$\iff K(\Lambda \circ \phi)\Lambda(\mathbb{F})(\lambda) = \top \iff \lambda \in \Gamma(K(\Lambda \circ \phi)\Lambda(\mathbb{F})). \quad \square$$

**Theorem 5.** *Let* $(X, p, q)$ *be pair of* $\top$*-convergence spaces and L be continuous. Then* $\lim^q$ *is p-topological iff q is p-topological.*

**Proof.** Let $q$ be $p$-topological. We check that $(X, \lim^p, \lim^q)$ satisfies $p$-**(LG)**. Obviously, we need only prove that $\lim^q \mathcal{F}(x) = \top$ implies $\lim^q \mathcal{U}_p^\alpha(\mathcal{F})(x) = \top$ for any $\alpha \neq \bot$.

Note that for any $\alpha \neq \bot$ and any $x \in X$ we have

$$\mathcal{U}_p^\alpha(x) = \bigwedge\{\mathcal{F}|\lim^p \mathcal{F}(x) = \top\} = \bigwedge\{\mathcal{F}|\mathcal{F} \geq \Lambda(\mathbb{F}), \mathbb{F} \xrightarrow{p} x\}$$
$$= \bigwedge\{\Lambda(\mathbb{F})|\mathbb{F} \xrightarrow{p} x\} \overset{\text{Lemma1}}{=} \Lambda(\bigcap \mathbb{F}\{|\mathbb{F} \xrightarrow{p} x\}) = \Lambda(\mathbb{U}_p(x)).$$

Let $\lim^q \mathcal{F}(x) = \top$ then $\mathcal{F} \geq \Lambda(\mathbb{F})$ for some $\mathbb{F} \xrightarrow{q} x$. It follows by $p$-**(TG)** that $\mathbb{U}_p(\mathbb{F}) \xrightarrow{q} x$ and

$$\mathcal{U}_p^\alpha(\mathcal{F}) \geq \mathcal{U}_p^\alpha(\Lambda(\mathbb{F})) = K\mathcal{U}_p^\alpha\Lambda(\mathbb{F}) = K(\Lambda \circ \mathbb{U}_p)\Lambda(\mathbb{F}) \overset{\text{Lemma2(1)}}{\geq} \Lambda(k\mathbb{U}_p\mathbb{F}) = \Lambda(\mathbb{U}_p(\mathbb{F})),$$

and so $\lim^q \mathcal{U}_p^\alpha(\mathcal{F})(x) = \top$ as desired.

Conversely, let $\lim^q$ be $p$-topological. We check that $(X, p, q)$ satisfies $p$-**(TG)**.

Assume that $\mathbb{F} \xrightarrow{q} x$. It follows by $p$-**(LG)** that

$$\lim^q \mathcal{U}_p^\top(\Lambda(\mathbb{F}))(x) = \lim^q K\mathcal{U}_p^\top \Lambda(\mathbb{F}) = \lim^q K(\Lambda \circ \mathbb{U}_p)\Lambda(\mathbb{F}) = \top,$$

and then $K(\Lambda \circ \mathbb{U}_p)\Lambda(\mathbb{F}) \geq \Lambda(\mathbb{G})$ for some $\mathbb{G} \xrightarrow{q} x$. By Lemma 2(2) we have

$$k\mathbb{U}_p\mathbb{F} = \Gamma(K(\Lambda \circ \mathbb{U}_p)\Lambda(\mathbb{F})) \supseteq \Gamma\Lambda(\mathbb{G}) = \mathbb{G}.$$

So, $\mathbb{U}_p(\mathbb{F}) = k\mathbb{U}_p\mathbb{F} \xrightarrow{q} x$ as desired. $\quad \square$

## 4. Lower and Upper $p$-Topological Modifications in $\top$-Convergence Spaces

In this section, we shall discuss the $p$-topological modification in $\top$-convergence spaces.

At first, we fix a lemma for later use. The proof is obvious, so we omit it.

**Lemma 3.** (1) *If* $(X, q)$ *is p-topological, then* $\mathbb{F} \xrightarrow{q} x$ *implies* $\mathbb{U}_p^n(\mathbb{F}) \xrightarrow{q} x$ *for any* $n \in \mathbb{N}$.
(2) *If* $(X, q)$ *is p-topological, then* $(X, q)$ *is p'-topological for any* $p \leq p'$.
(3) $(X, \iota)$ *is p-topological for any* $p \in \top(X)$.

*4.1. Lower p-Topological Modification*

Corollary 3 shows that $p$-topologicalness is preserved under supremum in the lattice $\top(X)$. Lemma 3(3) shows that the indiscrete space $(X, \iota)$ is $p$-topological for any $p \in \top(X)$. These two results make the following definition available.

**Definition 10.** *Let* $(X, p, q)$ *be a pair of* $\top$*-convergence spaces. Then there is a finest p-topological* $\top$*-convergence structure* $\tau_p q$ *on X which is coarser than q. The structure* $\tau_p q$ *is called the lower p-topological modification of q.*

The next theorem gives a direct characterization on lower $p$-topological modification.

**Theorem 6.** *Let $p, q \in \top(X)$. Then $\mathbb{F} \xrightarrow{\tau_p q} x \iff \exists n \in \mathbb{N}, \mathbb{G} \xrightarrow{q} x$ s.t. $\mathbb{F} \supseteq \mathbb{U}_p^n(\mathbb{G})$.*

**Proof.** Let $q'$ be defined as $\mathbb{F} \xrightarrow{q'} x \iff \exists n \in \mathbb{N}, \mathbb{G} \xrightarrow{q} x$ s.t. $\mathbb{F} \supseteq \mathbb{U}_p^n(\mathbb{G})$. We need only check that $\tau_p q = q'$.

It is obvious that $q' \in \top(X)$ and $q' \le q$. We prove that $q'$ is $p$-topological. Indeed, let $\mathbb{F} \xrightarrow{q'} x$. Then there exists $n \in \mathbb{N}, \mathbb{G} \xrightarrow{q} x$ such that $\mathbb{F} \supseteq \mathbb{U}_p^n(\mathbb{G})$. It follows that $\mathbb{U}_p(\mathbb{F}) \supseteq \mathbb{U}_p(\mathbb{U}_p^n(\mathbb{G})) = \mathbb{U}_p^{n+1}(\mathbb{G})$ and so $\mathbb{U}_p(\mathbb{F}) \xrightarrow{q'} x$, as desired. Thus $q'$ is $p$-topological.

Let $(X, r)$ be $p$-topological and $r \le q$. We prove below $r \le q'$. Indeed, let $\mathbb{F} \xrightarrow{q'} x$. Then there exists $n \in \mathbb{N}, \mathbb{G} \xrightarrow{q} x$ such that $\mathbb{F} \supseteq \mathbb{U}_p^n(\mathbb{G})$, and then $\mathbb{G} \xrightarrow{r} x$ by $q \le r$. Since $r$ is $p$-topological it follows by Lemma 3(1) we have $\mathbb{F} \supseteq \mathbb{U}_p^n(\mathbb{G}) \xrightarrow{r} x$. Thus $r \le q'$. $\square$

**Theorem 7.** *Let $f : (X, q) \longrightarrow (X', q')$ and $f : (X, p) \longrightarrow (X', p')$ be continuous function between $\top$-convergence spaces. Then $f : (X, \tau_p q) \longrightarrow (X', \tau_{p'} q')$ is also continuous.*

**Proof.** For any $\mathbb{F} \in \mathbb{F}_L^\top(X)$ and $x \in X$.

$$
\begin{aligned}
\mathbb{F} \xrightarrow{\tau_p q} x \implies & \exists n \in \mathbb{N}, \mathbb{G} \xrightarrow{q} x \text{ s.t. } \mathbb{F} \supseteq \mathbb{U}_p^n(\mathbb{G}) \\
\implies & \exists n \in \mathbb{N}, f^{\Rightarrow}(\mathbb{G}) \xrightarrow{q'} f(x) \text{ s.t. } f^{\Rightarrow}(\mathbb{F}) \supseteq f^{\Rightarrow}(\mathbb{U}_p^n(\mathbb{G})) \\
\implies & \exists n \in \mathbb{N}, f^{\Rightarrow}(\mathbb{G}) \xrightarrow{q'} f(x) \text{ s.t. } f^{\Rightarrow}(\mathbb{F}) \supseteq \mathbb{U}_{p'}^n(f^{\Rightarrow}(\mathbb{G})) \\
\implies & f^{\Rightarrow}(\mathbb{F}) \xrightarrow{\tau_{p'} q'} (f(x)),
\end{aligned}
$$

where the second implication holds for $f : (X, q) \longrightarrow (X', q')$ being continuous, and the third implication holds by $f : (X, p) \longrightarrow (X', p')$ being continuous and Proposition 3(1). $\square$

The next theorem shows that lower $p$-topological modification behaves reasonably well relative to final structures.

**Theorem 8.** *Let $\{(X_i, q_i, p_i)\}_{i \in I}$ be pairs of spaces in $\top$-CS and let $q$ be the final structure w.r.t. The sink $((X_i, q_i) \xrightarrow{f_i} X)_{i \in I}$ with $X = \cup_{i \in I} f_i(X_i)$. If $(X, p)$ is in $\top$-CS such that each $f_i : (X_i, p_i) \longrightarrow (X, p)$ is a continuous interior function, then $\tau_p q$ is the final structure w.r.t. the sink $((X_i, \tau_{p_i} q_i) \xrightarrow{f_i} X)_{i \in I}$.*

**Proof.** Let $s$ denote the final structure w.r.t. The sink $((X_i, \tau_{p_i} q_i) \xrightarrow{f_i} X)_{i \in I}$. Let $\mathbb{F} \in \mathbb{F}_L^\top(X)$ and $x \in X$. Then

$$
\begin{aligned}
\mathbb{F} \xrightarrow{s} x \implies & \exists i \in I, x_i \in X_i, f_i(x_i) = x, \mathbb{G}_i \xrightarrow{\tau_{p_i} q_i} x_i \text{ s.t. } f_i^{\Rightarrow}(\mathbb{G}_i) \subseteq \mathbb{F}, \text{ by Theorem 6} \\
\implies & \exists i \in I, x_i \in X_i, n \in \mathbb{N}, f_i(x_i) = x, \mathbb{H}_i \xrightarrow{q_i} x_i \text{ s.t. } \mathbb{U}_{p_i}^n(\mathbb{H}_i) \subseteq \mathbb{G}_i, f_i^{\Rightarrow}(\mathbb{G}_i) \subseteq \mathbb{F}, \text{Proposition 3(1)} \\
\implies & \exists i \in I, x_i \in X_i, n \in \mathbb{N}, f_i^{\Rightarrow}(\mathbb{H}_i) \xrightarrow{q} x \text{ s.t. } \mathbb{U}_p^n(f_i^{\Rightarrow}(\mathbb{H}_i)) \subseteq f_i^{\Rightarrow}(\mathbb{U}_{p_i}^n(\mathbb{H}_i)) \subseteq f_i^{\Rightarrow}(\mathbb{G}_i), f_i^{\Rightarrow}(\mathbb{G}_i) \subseteq \mathbb{F} \\
\implies & \exists i \in I, x_i \in X_i, n \in \mathbb{N}, f_i^{\Rightarrow}(\mathbb{H}_i) \xrightarrow{q} x \text{ s.t. } \mathbb{U}_p^n(f_i^{\Rightarrow}(\mathbb{H}_i)) \subseteq \mathbb{F} \\
\implies & \mathbb{F} \xrightarrow{\tau_p q} x.
\end{aligned}
$$

Conversely,

$$\mathbb{F} \xrightarrow{\tau_p q} x \implies \exists n \in \mathbb{N}, \mathbb{G} \xrightarrow{q} x \text{ s.t. } \mathbb{U}_p^n(\mathbb{G}) \subseteq \mathbb{F}$$

$$\implies \exists i \in I, x_i \in X_i, f_i(x_i) = x, n \in \mathbb{N}, \mathbb{H}_i \xrightarrow{q_i} x_i \text{ s.t. } f_i^\Rightarrow(\mathbb{H}_i) \subseteq \mathbb{G}, \mathbb{U}_p^n(\mathbb{G}) \subseteq \mathbb{F}$$

$$\implies \exists i \in I, x_i \in X_i, f_i(x_i) = x, n \in \mathbb{N}, \mathbb{H}_i \xrightarrow{q_i} x_i \text{ s.t. } \mathbb{U}_p^n(f_i^\Rightarrow(\mathbb{H}_i)) \subseteq \mathbb{U}_p^n(\mathbb{G}), \mathbb{U}_p^n(\mathbb{G}) \subseteq \mathbb{F}$$

$$\implies \exists i \in I, x_i \in X_i, f_i(x_i) = x, n \in \mathbb{N}, \mathbb{H}_i \xrightarrow{q_i} x_i \text{ s.t. } f_i^\Rightarrow(\mathbb{U}_{p_i}^n(\mathbb{H}_i)) \subseteq \mathbb{U}_p^n(f_i^\Rightarrow(\mathbb{H}_i)) \subseteq \mathbb{F}$$

$$\implies \exists i \in I, x_i \in X_i, f_i(x_i) = x, n \in \mathbb{N}, \mathbb{U}_{p_i}^n(\mathbb{H}_i) \xrightarrow{\tau_{p_i} q_i} x_i \text{ s.t. } f_i^\Rightarrow(\mathbb{U}_{p_i}^n(\mathbb{H}_i)) \subseteq \mathbb{F}$$

$$\implies \mathbb{F} \xrightarrow{s} x,$$

where the fourth implication uses Proposition 3(2). □

The following corollary shows that lower $p$-topological modification behaves reasonably well relative to infimum in the lattice $\top(X)$.

**Corollary 4.** *Let $\{q_i | i \in I\} \subseteq \top(X)$, $p \in \top(X)$ and $q = \inf\{q_i | i \in I\}$. Then $\tau_p q = \inf\{\tau_p q_i | i \in I\}$.*

At last, we give the notion of topological modification. By Corollary 3, it is observed that topologicalness is preserved under supremum in the lattice $\top(X)$. Since the indiscrete space is topological, the following notion is available.

**Definition 11.** *Let $(X, q)$ be a $\top$-convergence space. Then there exists a finest topological $\top$-convergence structure $\tau q$ which is coarser than $q$. The structure $\tau q$ is called the topological modification of $(X, q)$. Indeed, $\tau q = \sup\{p | p \leq q \text{ and } p \text{ is topological}\}$.*

*4.2. Upper $p$-Topological Modification*

Note that for an arbitrary $p \in \top(X)$, the discrete space $(X, \delta)$ is generally not $p$-topological. Thus for a given $q \in \top(X)$, there may not exist $p$-topological $\top$-convergence structure on $X$ which is finer than $q$.

**Definition 12.** *Let $(X, p, q)$ be a pair of $\top$-convergence spaces. If there exists a coarsest $p$-topological $\top$-convergence structure $\tau^p q$ on $X$ which is finer than $q$, then it is called the upper $p$-topological modification of $q$.*

From Corollary 3 we easily conclude that the existence of $\tau^p q$ depends on the existence of a $p$-topological $\top$-convergence structure on $X$ which is finer than $q$. Additionally, note that $\tau_p \delta$ is the finest $p$-topological $\top$-convergence structure on $X$. Then it follows immediately that $\tau^p q$ exists if and only if $q \leq \tau_p \delta$. Using Theorem 6, this result can be stated as below.

**Theorem 9.** *Let $(X, p, q)$ be a pair of $\top$-convergence spaces. Then $\tau^p q$ exists if and only if $\mathbb{U}_p^n([x]_\top) \xrightarrow{q} x$ for all $x \in X$, $n \in \mathbb{N}$.*

**Proof.** For each $\mathbb{F} \in \mathbb{F}_L^\top(X)$ and each $x \in X$, by Theorem 6 we have

$$\mathbb{F} \xrightarrow{\tau_p \delta} x \iff \exists n \in \mathbb{N}, \mathbb{G} \xrightarrow{\delta} x \text{ s.t. } \mathbb{U}_p^n(\mathbb{G}) \subseteq \mathbb{F}.$$

*Necessity.* Let $\tau^p q$ exist. Then $q \leq \tau_p \delta$. It follows that for all $x \in X, n \in \mathbb{N}$

$$[x]_\top \xrightarrow{\delta} x \implies \mathbb{U}_p^n([x]_\top) \xrightarrow{\tau_p \delta} x \implies \mathbb{U}_p^n([x]_\top) \xrightarrow{q} x.$$

*Sufficiency.* Let $\mathbb{U}_p^n([x]_\top) \xrightarrow{q} x$ for all $x \in X, n \in \mathbb{N}$. Then for all $\mathbb{F} \in \mathbb{F}_L^\top(X)$ we have

$$
\begin{array}{lll}
\mathbb{F} \xrightarrow{\tau_p \delta} x & \implies & \exists n \in \mathbb{N}, \mathbb{G} \xrightarrow{\delta} x \text{ s.t. } \mathbb{U}_p^n(\mathbb{G}) \subseteq \mathbb{F} \\
& \implies & \exists n \in \mathbb{N}, [x]_\top \subseteq \mathbb{G} \text{ s.t. } \mathbb{U}_p^n(\mathbb{G}) \subseteq \mathbb{F} \\
& \overset{\text{Proposition2(2)}}{\implies} & \exists n \in \mathbb{N}, \mathbb{U}_p^n(([x]_\top) \subseteq \mathbb{U}_p^n(\mathbb{G}) \text{ s.t. } \mathbb{U}_p^n(\mathbb{G}) \subseteq \mathbb{F} \\
& \implies & \exists n \in \mathbb{N} \text{ s.t. } \mathbb{U}_p^n(([x]_\top) \subseteq \mathbb{F} \\
& \implies & \mathbb{F} \xrightarrow{q} x.
\end{array}
$$

It follows that $q \leq \tau_p \delta$, which means that $\tau^p q$ exists. $\quad\square$

The next theorem gives a direct characterization on upper $p$-topological modification whenever it exists.

**Theorem 10.** *Let $(X, p, q)$ be a pair of $\top$-convergence spaces. If $\tau^p q$ exists, then $\mathbb{F} \xrightarrow{\tau^p q} x \iff \forall n \in \mathbb{N}, \mathbb{U}_p^n(\mathbb{F}) \xrightarrow{q} x$.*

**Proof.** Let $q'$ be defined as $\mathbb{F} \xrightarrow{q'} x \iff \forall n \in \mathbb{N}, \mathbb{U}_p^n(\mathbb{F}) \xrightarrow{q} x$.

(1) $q' \in \top(X)$.

(TC1) Let $x \in X$. Then by Theorem 9 we have $\mathbb{U}_p^n([x]_\top) \xrightarrow{q} x$ for all $n \in \mathbb{N}$, which means $[x]_\top \xrightarrow{q'} x$.
(TC2) It is obvious.

(2) $q \leq q'$. Indeed, let $\mathbb{F} \xrightarrow{q'} x$ then $\mathbb{F} = \mathbb{U}_p^0(\mathbb{F}) \xrightarrow{q} x$.

(3) $(X, q')$ is $p$-topological. Indeed, let $\mathbb{F} \xrightarrow{q'} x$. Then for any $n \in \mathbb{N}$ we have $\mathbb{U}_p^n(\mathbb{U}_p(\mathbb{F})) = \mathbb{U}_p^{n+1}(\mathbb{F}) \xrightarrow{q} x$, which means $\mathbb{U}_p(\mathbb{F}) \xrightarrow{q'} x$. Thus $(X, q')$ is $p$-topological.

(4) Let $(X, r)$ be $p$-topological and $q \leq r$. Then $q' \leq r$. Indeed, let $\mathbb{F} \xrightarrow{r} x$ then for any $n \in \mathbb{N}$, by Proposition 3(1) we have $\mathbb{U}_p^n(\mathbb{F}) \xrightarrow{r} x$ and so $\mathbb{U}_p^n(\mathbb{F}) \xrightarrow{q} x$ by $q \leq r$. That means $\mathbb{F} \xrightarrow{q'} x$.

(1)–(4) show that $q'$ is the coarsest $p$-topological $\top$-convergence structure on $X$ which is finer than $q$. Thus $\tau^p q = q'$. $\quad\square$

**Theorem 11.** *Let $f : (X, q) \longrightarrow (X', q')$ be a continuous function, and $f : (X, p) \longrightarrow (X', p')$ be an interior function between $\top$-convergence spaces. If $\tau^p q$ and $\tau^{p'} q'$ exist then $f : (X, \tau^p q) \longrightarrow (X', \tau^{p'} q')$ is also continuous.*

**Proof.** Let $\mathbb{F} \xrightarrow{\tau^p q} x$. Then $\forall n \in \mathbb{N}, \mathbb{U}_p^n(\mathbb{F}) \xrightarrow{q} x$. Because $f : (X, q) \longrightarrow (X', q')$ is a continuous function and $f : (X, p) \longrightarrow (X', p')$ is an interior function we have

$$\forall n \in \mathbb{N}, \mathbb{U}_{p'}^n(f^\Rightarrow(\mathbb{F})) \supseteq f^\Rightarrow(\mathbb{U}_p^n(\mathbb{F})) \xrightarrow{q'} f(x),$$

which means $f^\Rightarrow(\mathbb{F}) \xrightarrow{\tau^{p'} q'} f(x)$ as desired. $\quad\square$

The next theorem shows that the upper $p$-topological modification exhibits comparable behavior relative to initial structures.

**Theorem 12.** *Let $\{(X_i, q_i, p_i)\}_{i \in I}$ be pairs of spaces in $\top$-**CS** and $q$ be the initial structure w.r.t. The source $(X \xrightarrow{f_i} (X_i, q_i))_{i \in I}$. Let $(X, p)$ be in $\top$-**CS** such that each $f_i : (X, p) \longrightarrow (X_i, p_i)$ is continuous interior function. If $\tau^{p_i} q_i$ exists for all $i \in I$, then $\tau^p q$ exists and is the initial structure w.r.t. The source $(X \xrightarrow{f_i} (X_i, \tau^{p_i} q_i))_{i \in I}$.*

**Proof.** To prove $\tau^p q$ exists, it suffices, by Theorem 10, to show that $\mathbb{U}_p^n([x]_\top) \xrightarrow{q} x$ for any $x \in X, n \in \mathbb{N}$. Indeed, by the existence of $\tau^{p_i} q_i$ we have $\mathbb{U}_{p_i}^n([f_i(x)]) \xrightarrow{q_i} f_i(x)$ for any $i \in I, x \in X, n \in \mathbb{N}$. It follows by that each $f_i : (X, p) \longrightarrow (X_i, p_i)$ being a continuous interior function we get

$$f_i^{\Rightarrow}(\mathbb{U}_p^n([x]_\top) = \mathbb{U}_{p_i}^n(f_i^{\Rightarrow}([x]_\top)) = \mathbb{U}_{p_i}^n([f_i(x)]_\top) \xrightarrow{q_i} f_i(x),$$

which means $\mathbb{U}_p^n([x]_\top) \xrightarrow{q} x$ for any $x \in X, n \in \mathbb{N}$, i.e., $\tau^p q$ exists.

Let $s$ denote the initial structure on $X$ relative the source $(X \xrightarrow{f_i} (X_i, \tau^{p_i} q_i))_{i \in I}$. Then.

$$\mathbb{F} \xrightarrow{s} x \qquad \Longleftrightarrow \qquad \forall i \in I, f_i^{\Rightarrow}(\mathbb{F}) \xrightarrow{\tau^{p_i} q_i} f_i(x) \overset{\text{Theorem}10}{\Longleftrightarrow} \forall i \in I, \forall n \in \mathbb{N}, \mathbb{U}_{p_i}^n(f_i^{\Rightarrow}(\mathbb{F})) \xrightarrow{q_i} f_i(x)$$

$$\overset{\text{Proposition}3}{\Longleftrightarrow} \quad \forall i \in I, \forall n \in \mathbb{N}, f_i^{\Rightarrow}(\mathbb{U}_p^n(\mathbb{F})) \xrightarrow{q_i} f_i(x)$$

$$\Longleftrightarrow \quad \forall n \in \mathbb{N}, \mathbb{U}_p^n(\mathbb{F}) \xrightarrow{q} x \overset{\text{Theorem}10}{\Longleftrightarrow} \mathbb{F} \xrightarrow{\tau^p q} x. \quad \square$$

The following corollary shows that upper $p$-topological modification exhibits comparable behavior relative to supremum in the lattice $\top(X)$.

**Corollary 5.** *Let $\{q_i | i \in I\} \subseteq \top(X)$, $p \in \top(X)$ and $q = \sup\{q_i | i \in I\}$. If $\tau^p q_i$ exists for all $i \in I$, then $\tau^p q$ exists and $\tau^p q = \sup\{\tau^p q_i | i \in I\}$.*

## 5. Conclusions

In this paper, we discussed the $p$-topologicalness in $\top$-convergence spaces by a Fischer $\top$-diagonal condition and a Gähler $\top$-neighborhood condition, respectively. We proved that the $p$-topologicalness was preserved under the initial and final structures in the category $\top$-**CS**. As a straightforward conclusion, we further obtained that $p$-topologicalness was naturally preserved under the infimum and supremum in the lattice $\top(X)$. We also established the relationship between $p$-topologicalness in $\top$-convergence spaces and $p$-topologicalness in stratified $L$-generalized convergence spaces. Furthermore, we defined and studied the lower and upper $p$-topological modifications in $\top$-convergence spaces. In particular, we proved that the lower (resp., upper) $p$-topological modification exhibited comparable behavior relative to final (resp., initial) structures.

**Funding:** This work is supported by National Natural Science Foundation of China (No. 11801248, 11501278).

**Acknowledgments:** The author thanks the reviewers and the editor for their valuable comments and suggestions.

**Conflicts of Interest:** The author declares no conflict of interest.

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
