# Peer review of "p-Topologicalness—A Relative Topologicalness in ⊤-Convergence Spaces"

_mathematics, doi:10.3390/math7030228_

Round 1
Reviewer 1 Report
- The connection of the manuscript with Fuzzy Sets theory should be underlined in the Introduction (one paragraph before the end of it and after the sentence "This paper...")
- The word "References" should be added at the top of the list of References
- Is the footnote "Preprint submitted to Elsevier" at the bottom of Page 1 necessary?
Author Response
Response to Reviewer 1 Comments
Dear Prof
Thank you very much for your useful comments and valuable suggestions. We have revised the paper according to your report. We hope that meet the requirement!
For the convenience of reviewers, we use bold characters to write the revised parts.
Point 1: The connection of the manuscript with Fuzzy Sets theory should be underlined in the Introduction (one paragraph before the end of it and after the sentence "This paper...")
Response 1: We have pointed out that connection in the Introduction section, see page 1, line 21-23.
Point 2: The word "References" should be added at the top of the list of References.
Response 2: We have added it.
Point 3: Is the footnote "Preprint submitted to Elsevier" at the bottom of Page 1 necessary?
Response 3: We have deleted them.
Thanks a lot for your helpful suggestions again.
Best wishes!
Lingqiang Li

Reviewer 2 Report
The article is written in a good and clear form. Outputs are adequately presented.
Minor comments
- Numbered lists - Use each number on a new line (lines 45-47 and others)
- Please thoroughly check the formulas
- Modify some formulas to not overflow to the next line (line 103 and others)
Author Response
Response to Reviewer 2 Comments
Dear Prof
Thank you very much for your useful comments and valuable suggestions. We have revised the paper according to your report. We hope that meet the requirement!
For the convenience of reviewers, we use blue characters to write the revised parts.
Point 1: Numbered lists - Use each number on a new line (lines 45-47 and others).
Response 1: We have revised them. Please see
lines 44-49;
line 165-167;
line 280-281.
Point 2: Please thoroughly check the formulas.
Response 2: We have checked all formulas and revised one formula in Theorem 4.12. Please see, line 374.
Point 3: Modify some formulas to not overflow to the next line (line 103 and others).
Response 3: We have revised them. Please see
line 107;
line 175;
line 306;
line 235.
Thanks a lot for your helpful suggestions again.
Best wishes!
Lingqiang Li
